

# The role of heat shock proteins in HIV-1 pathogenesis: a systematic review investigating HSPs-HIV-1 correlations and interactions

Chyntia Tresna Nastiti[1], Noer Halimatus Syakdiyah[1], R.M. Firzha Hawari[1], Youdiil Ophinni[2,3] and Ni Luh Ayu Megasari[4,5]

[1] Postgraduate School, Universitas Airlangga, Surabaya, East Java, Indonesia
[2] Division of Clinical Virology, Graduate School of Medicine, Kobe University, Kobe, Japan
[3] Department of Environmental Coexistence, Center for Southeast Asian Studies, Kyoto University, Kyoto, Japan
[4] Immunology Program, Postgraduate School, Universitas Airlangga, Surabaya, East Java, Indonesia
[5] Institute of Tropical Disease, Universitas Airlangga, Surabaya, East Java, Indonesia

## ABSTRACT

**Background**. The human immunodeficiency virus (HIV) pandemic is a global health emergency. Studies suggest a connection between heat shock proteins (HSPs) and HIV-1 infection pathogenesis. This systematic review aims to summarize HSPs' role in HIV-1 infection pathogenesis.

**Materials and Methods**. A systematic literature search was undertaken across the National Library of Medicine (MEDLINE-PubMed), Science Direct, Web of Science, Scopus, SpringerLink, Sage, ProQuest, and Google Scholar databases, using related keywords to synthesize the HSPs' role in HIV-1 infection pathogenesis. This literature review was conducted according to Preferred Reporting Items for Systematic Reviews and Meta-Analyses (PRISMA) guidelines, and the protocol was registered in the Open Science Framework (OSF) database under DOI 10.17605/OSF.IO/VK3DJ.

**Results**. A database search revealed 3,332 articles, with 14 *in vitro* studies analysing the interaction between HSPs and HIV-1 across different cell types. HSPs are involved in HIV-1 infection through direct interactions and indirect responses to cellular stress, including HSP40, HSP70, HSPBP1, and HSP90. The study explores HSP interactions at various stages of the viral life cycle, including entry, uncoating, replication, transmission, and latency reactivation.

**Conclusion**. HSPs are crucial for the HIV lifecycle and immune response, offering the potential for new therapeutic strategies. Further research is needed to understand the clinical significance and target potential.

Corresponding author
Ni Luh Ayu Megasari,
ni.luh.ayu@pasca.unair.ac.id

# INTRODUCTION

Human immunodeficiency virus (HIV) is a retrovirus that targets CD4 T cells in particular, which infection may result in severe immunodeficiency characterized by the failure of host adaptive immune defense to combat opportunistic infections and malignancies (*Waymack*

*& Sundareshan, 2023*). Based on its genetic variation, HIV is categorized into two distinct types: HIV type 1 (HIV-1) and type 2 (HIV-2) (*Esbjörnsson et al., 2019*). Compared to HIV-2, which is primarily confined to West Africa, HIV-1 is more virulent and widespread and responsible for a pandemic that has been ongoing for over forty years. The HIV-1 pandemic is a significant public health emergency since the associated immunodeficiency allows a myriad of infectious diseases that may spread within and between different countries and geographic regions (*Esbjörnsson et al., 2019*; *Waymack & Sundareshan, 2023*).

Left untreated, HIV-1 insidiously replicates T cells and establishes latency, leading to chronic progression and severe disruption of the immune system, presenting as acquired immunodeficiency syndrome (AIDS) (*Justiz Vaillant & Gulick, 2023*). Around 40.4 million individuals have died from HIV, and 85.6 million have been infected ever since the beginning of the pandemic. At the end of 2022, 39 million people were living with HIV/AIDS worldwide. It is estimated that 0.7% (0.6%–0.8%) of adults between 15 and 49 are infected with HIV, even if the pandemic's prevalence and severity considerably vary between different countries and regions (*World Health Organization, 2023*).

The pathophysiology of HIV-1 is intricate, involving viral entry, replication, integration, and latency establishment. HIV-1 uses different strategies in targeting various immune cells, such as macrophages and T cells, and tissues, such as the mucosal gut barrier and brain tissue (*Vidya Vijayan et al., 2017*; *Masenga et al., 2023*). The virus has to navigate through the complexities of cellular machinery to support its life cycle, hijacking host physiological processes while evading restriction factors with its accessory proteins, such as the HIV-1 Vpr protein that hijacks E3 ubiquitin ligases, Vif protein that antagonizes APOBEC3G-mediated hypermutation, and Vpu that counteracts viral release inhibition by CD317/Tetherin (*Simon, Bloch & Landau, 2015*; *Yamashita & Engelman, 2017*). The sheer number and amount of details in HIV-1-host interactions underscore the decades worth of scientific efforts to understand the cellular pathophysiology of HIV-1, and understanding these various viral-host processes that influence infection, replication, and latency is paramount to approach closer to the discovery of vaccine or cure for this virus (*Masenga et al., 2023*).

The management of HIV-1 infection mainly relies on antiretroviral therapy (ART), which typically includes three medications from two ARV drug classes (*Tyler & Peter, 2023*). ART is thought to be effective in suppressing viral replication and support the recovery of CD4 T-cell counts (*Bahemana et al., 2020*; *Megasari & Wijaksana, 2023*). However, despite the availability of ART, the management of HIV-1 infection currently faces several challenges, including the emergence of drug resistant strains which might reduce the effectiveness of various ARV drug classes. HIV drug resistance might affect both ART-naïve and experienced individuals (*Megasari et al., 2019b*; *Megasari et al., 2019a*; *Khairunisa et al., 2020b*; *Khairunisa et al., 2020a*; *Khairunisa et al., 2020c*; *Khairunisa et al., 2023*; *World Health Organization, 2021*; *Bertagnolio et al., 2022*); therefore, exploring novel therapeutic strategies for the management of HIV-1 infection is urgent.

Heat shock proteins (HSPs) are ubiquitous and conserved family of proteins which maintains cell proteostasis and protects cells from stresses (*Iyer et al., 2021*; *Hu et al., 2022*; *Zhang & Yu, 2022*). They function as molecular chaperones and have been frequently

highlighted in cell physiological studies due to their multifaceted roles in many processes, including protein folding and refolding, complex assembly, and degradation (*Hu et al., 2022*; *Zhang & Yu, 2022*). HSPs are categorized based on molecular weight, where each member displays distinct functions and is implicated in various diseases like cancer and infection (*Hu et al., 2022*). In viral infection, HSPs have been suggested as one of the critical actors in antigen presentation and antiviral innate immunity response and have been implicated in several viral replication processes, including dengue, influenza, and hepatitis C virus (*Ujino et al., 2009*; *Manzoor et al., 2014*; *Taguwa et al., 2015*; *Wan et al., 2020*). A previous review study reported that several cellular HSPs interact with HIV-1 proteins and virions, and thus may be involved in the progression of HIV-1 infection, such as HSP40 and HIV-1 Nef protein to enhance viral translocation into the nucleus, HSP70 and HIV-1 virions as well as HIV-1 gag protein, and may be involved in the viral uncoating process, HSP90 blocks p-TEFb to suppress reactivation from latency. Other HSP family members appear to interact with HIV-1, but the mechanism or function remains unclear (*Iyer et al., 2021*).

HSPs have been known to catalyze cytokine secretion in antigen-presenting cells (*Moré, Breloer & Von Bonin, 2001*), but at the same time, HSPs also modulate different immune cells, such as T cells and regulatory T cells (*Hauet-Broere et al., 2006*; *Brenu et al., 2013*). Meanwhile, the involvement of HSPs in viral pathogen recognition, viral immune response modulation, and even direct interactions with viral proteins indicate that they are crucial to how an infection turns out (*Wan et al., 2020*; *Zhang & Yu, 2022*).

Recent studies have hinted at a significant connection between HSPs and HIV-1 infection pathogenesis (*Iyer et al., 2021*; *Zhang & Yu, 2022*). Given the significance of HSPs in cellular processes yet their enigmatic role in the framework of HIV-1 infection, a comprehensive review of their interplay is essential. Herein, we aim to delve into the current literature to clarify the role of HSPs in HIV-1 pathogenesis, offering insights that might pave the way to a deeper understanding of HIV-1-host interactions and possible novel therapeutic strategies.

## MATERIALS & METHODS

### Registration

This study was performed according to the Preferred Reporting Items for Systematic Review and Meta-analysis (PRISMA) guidelines, and the protocol was recorded in the Open Science Framework (OSF) database under following registration DOI: 10.17605/OSF.IO/VK3DJ.

### Search strategy

This systematic review was conducted by looking for among eight electronic databases, including the Scopus, National Library of Medicine (MEDLINE-PubMed), Web of Science, Science Direct, SpringerLink, Sage, ProQuest, and Google Scholar, using Medical Subject Headings (MeSH) terms and various combinations of the following keywords: *heat shock proteins* or *HSP*, *stress protein*, and *human immunodeficiency virus-1* or *HIV-1*. The databases searched is studies conducted from January 2013 to May 2024. The structured search strategy was designed to identify any published article in English that analysed the

role of HSPs in HIV-1 infection, and described the evaluation instruments used in *in vitro* and *in vivo* studies. To prioritize the quality of evidence, we refrained from searching into grey literature and preprints. We also did not directly contact the investigators or try to identify unpublished data.

## Inclusion and exclusion criteria

Studies with full-text papers were examined for inclusion and exclusion criteria. The inclusion criteria for this study included any relevant *in vitro* and *in vivo* studies that report the correlation or role of HSP in HIV-1 infection. Non-full text articles, non-English articles, duplicate articles in human studies, and irrelevant data that did not meet eligibility criteria, such as review articles, meta-analysis articles, editorials/letters, and congress abstracts, were excluded.

## Study selection and data extraction

Two authors (NHS and CTN) independently screened the articles and selected relevant studies based on the inclusion and exclusion criteria. The first screening is based on the information stated in the title and abstract. The second screening was based on reports on full-text articles. Articles were entered into this study if they met all eligibility criteria the following these two screening steps. Disagreements between the two authors were cleared up through discussion with a third author (NLAM). The enrolled data from selected articles were inserted in the data extraction table. We extracted the title of articles, author and year of publication, country of origin, study design, animal model or cell type used, assay methods, HSP type, and the outcome for each included study.

## Quality assessment

The process of quality assessment, evaluation, and resolution of discrepancies is conducted and evaluated by the same authors as in the Study Selection and Data Extraction section. The included studies were submitted to assess the inherent study quality based on the ToxRTool, which contains five domains: (1) Test substance identification, (2) test system characterization, (3) study design description, (4) Study results documentation, and (5) plausibility of study design and data (*ToxRTool Toxicological data reliability assessment tool, 2023*). Studies are deemed reliable without restrictions (total score 15–18), reliable with restrictions (total score 11–14), or not-reliable (total score < 11).

# RESULTS

## Search and screening results

We found 119 articles in PubMed, 204 in Scopus, 1,082 in Science Direct, 101 in Web of Science, 1,733 in SpringerLink, 13 in ProQuest, 17 in Google Scholar, and 63 in Sage. In total, 3,332 articles were identified. After deleting duplicate reports, we continued with the reading of 3,075 titles and abstracts, whereafter 37 articles were identified to read in full following exclusion of studies which are out-of-scope or not adhering to inclusion criteria. Finally, 33 studies were assessed for eligibility, in which 14 studies were included in this review (*Joshi et al., 2013*; *Urano, Morikawa & Komano, 2013*;

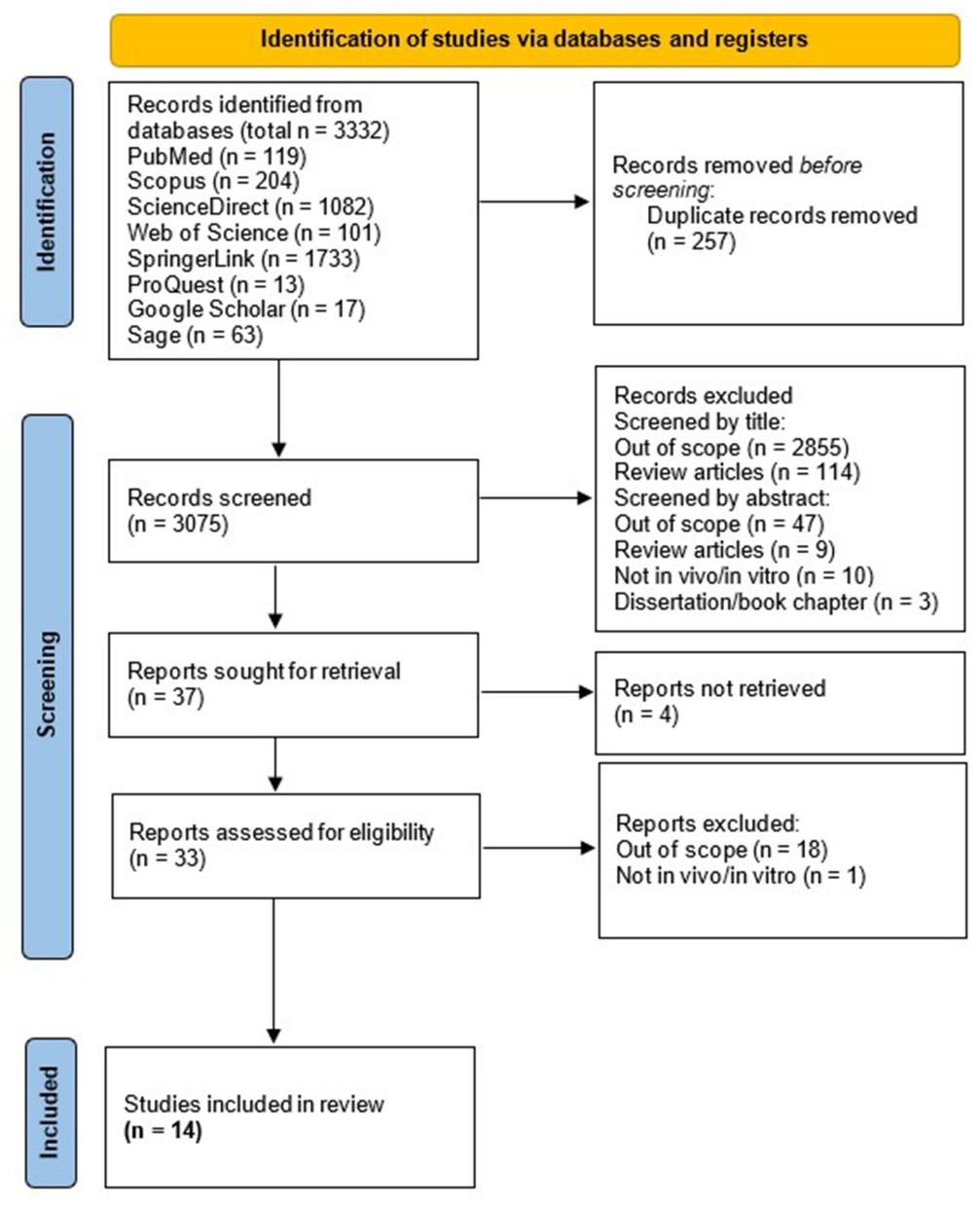

**Figure 1  PRISMA Flow Diagram of the studies selection.**

*Mercier et al., 2013*; *Anderson et al., 2014*; *Xia et al., 2015*; *Chaudhary et al., 2016*; *Joshi, Maidji & Stoddart, 2016*; *Painter, Zaikos & Collins, 2017*; *Mbonye et al., 2018*; *Ko et al., 2019*; *Priyanka Wadhwa et al., 2020*; *Chand, Iyer & Mitra, 2021*; *Iyer, Mitra & Mitra, 2022*; *Chand et al., 2023*). The article collection, screening, and eligibility assessment process is shown in Fig. 1.
## Characteristics

The included studies demonstrated a variety of research conducted globally from January 2013 to December 2023, focusing on the role of HSPs and their interaction with HIV-1. The studies originate from different countries, including Japan, the USA, Australia, the UK, China, Taiwan, and India, and have been published in peer-reviewed journals. The characteristics of the included studies are presented in Table 1.

## HSP and HIV-1 correlations

Table 2 shows the 14 *in vitro* studies that were included of this review, which provide a comprehensive analysis of the interaction between various HSPs and HIV-1 across different cell types and a range of assay methods, including PCR, RT-PCR, ELISA, Western blotting, and immunoprecipitation, among others. Two studies focused on the HSP40 class (*Ko et al., 2019*; *Chand et al., 2023*), two studies on HSP40 and HSP70 (*Urano, Morikawa & Komano, 2013*; *Chand, Iyer & Mitra, 2021*), two studies on HSP70 (*Xia et al., 2015*; *Priyanka Wadhwa et al., 2020*), six studies on HSP90 (*Joshi et al., 2013*; *Mercier et al., 2013*; *Anderson et al., 2014*; *Joshi, Maidji & Stoddart, 2016*; *Painter, Zaikos & Collins, 2017*; *Mbonye et al., 2018*), and two studies on HSPBP1 (*Chaudhary et al., 2016*; *Iyer, Mitra & Mitra, 2022*).

The selected studies employed various cell lines and primary cells, each providing unique insights into the complex interactions between HSPs and HIV-1. The cell lines used include human embryonic kidney (HEK-293T) (*Urano, Morikawa & Komano, 2013*; *Mercier et al., 2013*; *Chaudhary et al., 2016*; *Ko et al., 2019*; *Iyer, Mitra & Mitra, 2022*; *Chand et al., 2023*), human lung epithelial A549 (*Ko et al., 2019*), human monocytic THP-1 (*Ko et al., 2019*), and human T lymphocitic Jurkat cells (*Joshi et al., 2013*; *Anderson et al., 2014*; *Chaudhary et al., 2016*; *Joshi, Maidji & Stoddart, 2016*; *Mbonye et al., 2018*; *Iyer, Mitra & Mitra, 2022*). Additionally, the following primary cell lines for *ex vivo* analyses were also utilized: primary rat cortical neurons (*Xia et al., 2015*), human fetal brain-derived astrocytes (*Priyanka Wadhwa et al., 2020*), human peripheral blood mononuclear cells (PBMCs), and human hematopoietic stem cells (*Chaudhary et al., 2016*; *Painter, Zaikos & Collins, 2017*; *Chand, Iyer & Mitra, 2021*). These diverse cellular models allowed for a comprehensive exploration of the roles of different HSP families in various aspects of HIV-1 replication, infectivity, and latency.

This systematic review reveals that HSP40 family members play a critical role in HIV-1 replication and infectivity, with various studies indicating that the downregulation or silencing of specific HSP40 isoforms leads to reduced HIV-1 replication and promoter activity (*Urano, Morikawa & Komano, 2013*; *Ko et al., 2019*; *Chand, Iyer & Mitra, 2021*; *Chand et al., 2023*). Conversely, overexpression of certain HSP40 isoforms increases HIV-1 LTR promoter activity in a dose dependent manner (*Chand et al., 2023*). The HSP70 family is also implicated, with findings suggesting that these proteins are implicated in the inherent immunity against HIV-1, with their expression levels modulated during different infection stages (*Urano, Morikawa & Komano, 2013*; *Xia et al., 2015*; *Chand, Iyer & Mitra, 2021*). Mortalin, an HSP70 family member, is shown to interact with HIV-1 Tat protein, mitigating its availability for cellular toxicity (*Priyanka Wadhwa et al., 2020*).

**Table 1  Characteristics of the included studies.**

| No. | Publication Title | Authors | Journal | Year | Country of Origin | Citation |
|---|---|---|---|---|---|---|
| 1 | Novel role of HSP40/DNAJ in the regulation of HIV-1 replication | Urano E, Morikawa Y, Komano J | Journal of Acquired Immune Deficiency Syndromes | 2013 | Japan | *Urano, Morikawa & Komano (2013)* |
| 2 | Heat shock protein 90AB1 and hyperthermia rescue infectivity of HIV with defective cores | Joshi P, Sloan B, Torbett BE, Stoddart CA | Journal of Virology | 2013 | USA | *Joshi et al. (2013)* |
| 3 | The microvesicle component of HIV-1 inocula modulates dendritic cell infection and maturation and enhances adhesion to and activation of T lymphocytes | Mercier SK, Donaghy H, Botting RA, Turville SG, Harman AN, Nasr N, Ji H, Kusebauch U, Mendoza L, Shteynberg D, Sandgren K, Simpson RJ, Moritz RL, & Cunningham AL | PLOS ONE Pathogens | 2013 | Australia | *Mercier et al. (2013)* |
| 4 | Heat shock protein 90 controls HIV-1 reactivation from latency | Anderson I, Low JS, Weston S, Weinberger M, Zhyvoloup A, Labokha AA, Corazza G, Kitson RA, Moody CJ, Marcello A, Fassati A | Proceedings of the National Academy of Sciences | 2014 | UK | *Anderson et al. (2014)* |
| 5 | Curcumin increases HSP70 expression in primary rat cortical neuronal apoptosis induced by gp120 V3 loop peptide | Xia C, Cai Y, Li S, Yang J, Xiao G | Neurochemical Research | 2015 | China | *Xia et al. (2015)* |
| 6 | HSP70 binding protein 1 (HspBP1) suppresses HIV-1 replication by inhibiting NF-$\kappa$B mediated activation of viral gene expression | Chaudhary P, Khan SZ, Rawat P, Augustine T, Raynes DA, Guerriero V, Mitra D | Nucleic Acids Research | 2016 | India | *Chaudhary et al. (2016)* |
| 7 | Inhibition of heat shock protein 90 prevents HIV rebound | Joshi P, Maidji E, Stoddart CA | Journal of Biological Chemistry | 2016 | USA | *Joshi, Maidji & Stoddart (2016)* |
| 8 | Quiescence promotes latent HIV infection and resistance to reactivation from latency with histone deacetylase inhibitors | Painter MM, Zaikos TD, Collins KL | Journal of Virology | 2017 | USA | *Painter, Zaikos & Collins (2017)* |

Nastiti et al. (2024), *PeerJ*, DOI 10.7717/peerj.18002

**Table 1** (*continued*)

| No. | Publication Title | Authors | Journal | Year | Country of Origin | Citation |
|-----|-------------------|---------|---------|------|-------------------|----------|
| 9 | Cyclin-dependent kinase 7 (CDK7)-mediated phosphorylation of the CDK9 activation loop promotes P-TEFb assembly with Tat and proviral HIV reactivation. | Mbonye U, Wang B, Gokulrangan G, Shi W, Yang S, Karn J | Journal of Biological Chemistry | 2018 | USA | *Mbonye et al. (2018)* |
| 10 | Interference of DNAJB6/MRJ isoform switch by Morpholino inhibits replication of HIV-1 and RSV | Ko SH, Liau YJ, Chi YH, Lai MJ, Chiang YP, Lu CY, Chang LY, Tarn WY, Huang LM | Molecular Therapy-Nucleic Acids | 2019 | Taiwan | *Ko et al. (2019)* |
| 11 | Novel role of mortalin in attenuating HIV-1 Tat-mediated astrogliosis | Priyanka, Wadhwa R, Chaudhuri R, Nag TC, Seth P | Journal of Neuroinflammation | 2020 | India | *Priyanka et al. (2020)* |
| 12 | Comparative analysis of differential gene expression of HSP40 and HSP70 family isoforms during heat stress and HIV-1 infection in T-cells | Chand K, Iyer K, Mitra D | Cell Stress and Chaperones | 2021 | India | *Chand, Iyer & Mitra (2021)* |
| 13 | Identification of 5′ upstream sequence involved in HSPBP1 gene transcription and its downregulation during HIV-1 infection | Iyer K, Mitra A, Mitra D | Virus Research | 2022 | India | *Iyer, Mitra & Mitra (2022)* |
| 14 | DNAJB8 facilitates autophagic-lysosomal degradation of viral Vif protein and restricts HIV-1 virion infectivity by rescuing APOBEC3G expression in host cells | Chand K, Barman MK, Ghosh P, Mitra D | FASEB Journal | 2023 | India | *Chand et al. (2023)* |

Nastiti et al. (2024), *PeerJ*, DOI 10.7717/peerj.18002

**Table 2  Summary of included studies.**

| Citation | Study design | Animal/cell type | Assay methods | HSP class type | HSP-HIV-1 correlation |
|---|---|---|---|---|---|
| Ko et al. (2019) | *in vitro* | HEK-293T cells, human lung epithelial A549 cells, Human monocytes, THP-1 | PCR, RT-PCR, Southern Blotting, Immunoblotting | HSP40 | Downregulation of the MRJ-L level (an HSP40 family member) reduced HIV-1 replication. |
| Chand et al. (2023) | *in vitro* | HEK-293T cell line (human embryonic kidney) | qRT-PCR, cycloheximide chase assay | HSP40 | Silencing of DNAJA3, DNAJB1, DNAJB7, DNAJC4, DNAJC5B, DNAJC5G, DNAJC6, DNAJC22, and DNAJC30 was found to inhibit HIV-1 LTR promoter activity, whereas overexpression of these HSPs enhanced LTR promoter activity in a dose-dependent manner. Knockdown of DNAJB3, DNAJB6, DNAJB8, and DNAJC5 significantly enhanced promoter activity, while overexpression of them significantly decreased LTR promoter activity in a dose-dependent manner. By downregulating the expression of HIV-1 Vif protein, DNAJB8 was found to limit the infectivity of progeny virions. The infectivity of progeny virion particles was determined by the expression level of DNAJB8 protein during early and late stages of HIV-1 infection in T cells. |

Nastiti et al. (2024), *PeerJ*, DOI 10.7717/peerj.18002

**Table 2** (*continued*)

| Citation | Study design | Animal/cell type | Assay methods | HSP class type | HSP-HIV-1 correlation |
|---|---|---|---|---|---|
| *Urano, Morikawa & Komano (2013)* | *in vitro* | HEK-293T cell | ELISA, western blotting, immunoprecipitation, immunofluorescence assay, RT-PCR | HSP40 and HSP70 | HSP40 family limited HIV-1 production through activation of HSP70. HSP40A1 and B6 were able to form a complex with HSP70 that possessed a novel antiviral function specific to HIV-1 infection. HSP70-HSP40 complex conferred inherent immunity against HIV-1. HSP40A1, B1, and B6 were expressed in T cells, and HSP40B6 expression was upregulated by IFN-$\alpha$. |
| *Chand, Iyer & Mitra (2021)* | *in vitro* | CEM-GFP (reporter T-cell line) and human PBMC | qRT-PCR | HSP40 and HSP70 | The majority of HSP40 isoforms were upregulated during the early phase of HIV-1 infection (days 1-5) up to the peak of infection (day 7), except for the DNAJB6a isoform, which was significantly downregulated on day 7. On the other hand, all HSP70 isoforms were downregulated during the early phase of infection. As the infection progressed towards the peak and late phases ($\geq$ day 7), some HSP70 isoforms such as HSPA1L, HSPA2, HSPA5, HSPA9, HSPA12, HSPA13 showed an increased expression. |
| *Xia et al. (2015)* | *in vitro* | Primary rat cortical neuron cell culture | Western blotting, qRT-PCR | HSP70 | The expression level of HSP70 protein was lower in cells treated with HIV-1 gp120 V3 loop peptide in dose-dependent manner. |
| *Priyanka Wadhwa et al. (2020)* | *in vitro* | Human fetal brain-derived astrocytes | RT-PCR, western blotting, immunofluorescence, immunocytochemistry, immunohistochemistry | HSP70 | Mortalin (a member of HSP70 family) level was downregulated in Tat transfected astrocytes. Mortalin directly bind to Tat in astrocytes and degraded Tat, making Tat unavailable for cellular toxicity. Overexpression of mortalin in the presence of Tat reduced the cytotoxic effect of Tat in astrocytes. |

Nastiti et al. (2024), *PeerJ*, DOI 10.7717/peerj.18002

**Table 2** (*continued*)

| Citation | Study design | Animal/cell type | Assay methods | HSP class type | HSP-HIV-1 correlation |
|---|---|---|---|---|---|
| *Joshi et al. (2013)* | *in vitro* | Jurkat E6-1 cells | Western blot | HSP90 | HSP90AB1 was found incorporated within virions, located within the virions but outside of the virus core. HIV with mutations in the capsid showed the ability to infect target cells that express HSP90AB1. Inhibition of HSP90AB1 resulted in the virus being non-infectious. |
| *Mercier et al. (2013)* | *in vitro* | MDDC, HEK-293T cells | Flowcytometry, western blot, qPCR | HSP90 | HSPs 90$\alpha$ and $\beta$, the major cellular HSPs were found expressed in HIV-1 microvesicles (MVs). Both HSP90$\alpha$ and $\beta$ in the MVs showed activity to induce maturation of dendritic cells and enhance ICAM-1, with HSP90b being the most potent. Increased dendritic cell maturation led to increased levels of HIV-1 transfer to T lymphocytes, facilitating the progress of HIV-1 infection. |
| *Anderson et al. (2014)* | *in vitro* | J-Lat A2 cells | Flowcytometry, Western blotting | HSP90 | Inhibition of HSP90 resulted in decreased reactivation of HIV-1 through the indirect inhibition of the PKC pathway activation, as well as the inhibition of IKK activation, degradation of IkB $\alpha$, and RelA/p65 nuclear translocation in the NF-$\kappa$B pathway. Meanwhile, reactivation of HIV-1 through the MEK/MAPK/P-TEFb pathway did not involve HSP90. |
| *Joshi, Maidji & Stoddart (2016)* | *in vitro* | IV-infected 8E5/LAV cells and ACH-2 cells and uninfected Jurkat E6-1 cells | Flowcytometry, Infectious Units per Million Cells (IUPM) Assay, Cell Viability Assay, qPCR, ELISA, Western Blotting, Electroporation, Immunoprecipitation (IP) and Co-IP | HSP90 | HSP90 was required to reactivate HIV transcription in persistent virus reservoirs. HSP90 inhibition suppressed HIV transcription in persistently infected cells, thus preventing rebound. |

**Table 2** (*continued*)

| Citation | Study design | Animal/cell type | Assay methods | HSP class type | HSP-HIV-1 correlation |
|---|---|---|---|---|---|
| *Painter, Zaikos & Collins (2017)* | *in vitro* | human-hematopoietic stem and progenitor cells (HP-SCs) | Flow cytometry, ELISA, Western Blot | HSP90 | HSP90 inhibition led to a significant increase in quiescent HPSCs, where HIV-1 provirus was latent and less activated in this cell type compared to the differentiated HPSCs. Quiescent HSPCs were resistant to reactivation by histone deacetylase inhibitors or P-TEFb activation but were susceptible to reactivation by protein kinase C (PKC) agonists and TNF-$\alpha$ stimulation. |
| *Mbonye et al. (2018)* | *in vitro* | Jurkat 2D10 T-cell line | Western blot, flow cytometry | HSP90 | HSP90 inhibition suppressed HIV transcription through degradation of CycT1 and therefore P-TEFb was not generated and prevent reactivation of latent HIV in primary T cell. |
| *Chaudhary et al. (2016)* | *in vitro* | Jurkat (CD4+ human T cell line), HEK-293T (human embryonic kidney cell line), human PBMC | luciferase assay, immunoblotting and immuno-precipitation assays, qRT-PCR, Electrophoretic mobility shift assay (EMSA), Chromatin-immunoprecipitation (ChIP) | HSPBP1 | HspBP1 interacted with HSP40 and HSP70 to inhibit the increase in LTR promoter activity and LTR-driven gene expression, respectively. Furthermore, HspBP1 independently suppressed LTR activity in the presence of Tat by inhibiting the NF-$\kappa$B pathway and reducing the interaction between LTR and p50/p65 proteins, leading to the inhibition of viral transcription in T cells in a dose-dependent manner. |
| *Iyer, Mitra & Mitra (2022)* | *in vitro* | HEK-293T and Jurkat cells | ELISA, qRT-PCR, immunoblotting, immunoprecipitation assay | HSPBP1 | HSPBP1 was downregulated during HIV-1 infection in T-cells. HIV-1 suppressed the HSPBP1 promoter activity. HIV-1 Tat bonded directly to the promoter or indirectly interacted with Sp1, where amino acids 32–48 formed the potential repressive motif. By suppressing HSPBP1 through its promoter (in 5′ upstream region of HSPBP1 gene), Tat neutralised the repressive effect of HSPBP1 on LTR-driven gene expression thereby creating an environment conducive for successful HIV-1 infection. |

Studies on the HSP90 family reveal its incorporation into HIV-1 virions and its important role in HIV-1 core stability and uncoating (*Joshi et al., 2013*). HSP90 inhibition suppresses HIV-1 transcription in persistently infected cells, preventing viral rebound (*Anderson et al., 2014*; *Joshi, Maidji & Stoddart, 2016*). Furthermore, HSP90 is required for the viral transmission and reactivation of latent HIV-1 in certain cell types, with its inhibition leading to increased quiescence and resistance to reactivation by specific agents (*Mercier et al., 2013*; *Painter, Zaikos & Collins, 2017*; *Mbonye et al., 2018*). Lastly, HSPBP1 interreacted with HSP40 and HSP70, suppressing HIV-1 LTR activity and viral transcription in T cells, with its expression being downregulated during the course of HIV-1 infection (*Chaudhary et al., 2016*; *Iyer, Mitra & Mitra, 2022*).

### Role of HSP in HIV infection pathogenesis

Table 3 describes the complex and varied interactions between HSPs and HIV-1 at different stages of the viral life cycle, starting from entry (*Xia et al., 2015*; *Priyanka Wadhwa et al., 2020*; *Chand et al., 2023*), uncoating (*Joshi et al., 2013*), replication (*Urano, Morikawa & Komano, 2013*; *Chaudhary et al., 2016*; *Priyanka Wadhwa et al., 2020*; *Iyer, Mitra & Mitra, 2022*), viral transmission, (*Mercier et al., 2013*) to latency reactivation (*Anderson et al., 2014*; *Joshi, Maidji & Stoddart, 2016*; *Painter, Zaikos & Collins, 2017*; *Mbonye et al., 2018*).

### Quality of the included studies

Table 4 shows that 11 studies out of 24 analyzed studies have scored over 15 points and were therefore considered highly reliable and relevant to their intended purpose (*Joshi et al., 2013*; *Mercier et al., 2013*; *Anderson et al., 2014*; *Xia et al., 2015*; *Chaudhary et al., 2016*; *Joshi, Maidji & Stoddart, 2016*; *Ko et al., 2019*; *Priyanka Wadhwa et al., 2020*; *Chand, Iyer & Mitra, 2021*; *Iyer, Mitra & Mitra, 2022*; *Chand et al., 2023*). In addition to these 11 studies, three others were considered reliable but with some restrictions. This means that although these studies had the potential to be useful for their intended purpose, they were limited in some way and, therefore, scored between 11 and 14 (*Urano, Morikawa & Komano, 2013*; *Painter, Zaikos & Collins, 2017*; *Mbonye et al., 2018*). Overall, these findings suggest that the majority of studies available can be relied upon for their quality and relevance without restrictions.

## DISCUSSION

Our review shows that, in the molecular process for HIV-1 infection, HSPs role as critical molecular chaperone implicated in different stages of viral infection. They play a role *via* direct interactions with viral proteins and indirect responses to cellular stress induced by viral infection (*Neckers & Tatu, 2008*). Such functions were exhibited by several HSP families studied in articles included in this review, particularly HSP40, HSP70, HSPBP1, and HSP90.

The HSP40 family, the most studied among all HSP isoforms, is involved in the viral entry and replication stages of HIV-1. The HSP40 family member B6/DNAJB6 is divided into two distinct isoforms based on the length of exons: the MRJ-L (large) and the MRJ-S (small)

Nastiti et al. (2024), *PeerJ*, DOI 10.7717/peerj.18002

**Table 3  The role of different HSP classes in HIV infection pathogenesis.**

| HSP Class | HSP family/isoform | HSP interaction effects | HIV infection cycle stage | Citation |
|---|---|---|---|---|
| HSP40 | DNAJB6/MRJ | Support HIV-1 replication | Replication | *Ko et al. (2019)* |
| HSP40 | DNAJA, DNAJB, DNAJC | Inhibit HIV-1 infection: DNAJB3, DNAJB6, DNAJB8, DNAJC5 support HIV-1 infection: DNAJA3, DNAJB1, DNAJB7, DNAJC4, DNAJC5B, DNAJC5G, DNAJC6, DNAJC22, DNAJC30 | Viral entry | *Chand et al. (2023)* |
| HSP40 and HSP70 | DNAJB, DNAJC, HSPA | Upregulated in early stage of HIV-1 infection: DNAJB3, DNAJB7, DNAJB8, DNAJB13, DNAJC5, DNAJC5B, DNAJC5G, DNAJC6, DNAJC30; downregulated in early stage of HIV-1 infection: DNAJB6a, HSPA1A, HSPA1B, HSPA6; upregulated in late stage of HIV-1 infection: HSPA1L, HSPA2, HSPA5, HSPA9, HSPA12, HSPA13; downregulated in late stage of HIV-1 infection: HSPA1A, HSPA1B, HSPA6, HSPA14. | Viral entry | *Chand, Iyer & Mitra (2021)* |
| HSP40 and HSP70 | HSP40A1, HSP40B1, HSP40B6, HSP40C5, HSP70 | Inhibit HIV-1 replication: HSP40A1, HSP40B1, HSP40B6, HSP40C5; bind to HSP70: HSP40A1 and HSP40B6. | Replication | *Urano, Morikawa & Komano (2013)* |
| HSP70 | – | Decreased by HIV-1 infection | Viral entry | *Xia et al. (2015)* |
| HSP70 | Mitochondrial HSP70 (mtHSP70)/Mortalin | Decreased by HIV-1 infection, lower cytotoxic effect | Replication | *Priyanka Wadhwa et al. (2020)* |
| HSP90 | HSP90$\beta$/HSP90AB1 | Increased by HIV-1 infection | Uncoating | *Joshi et al. (2013)* |
| HSP90 | HSP90$\alpha$ and HSP90$\beta$ | Increased by HIV-1 infection | Viral transmission | *Mercier et al. (2013)* |
| HSP90 | – | Suppress HIV-1 reactivation from latency | Latency | *Joshi, Maidji & Stoddart (2016)* |
| HSP90 | – | Suppress HIV-1 reactivation from latency | Latency | *Anderson et al. (2014)* |
| HSP90 | – | Suppress HIV-1 reactivation from latency | Latency | *Painter, Zaikos & Collins (2017)* |
| HSP90 | – | Suppress HIV-1 reactivation from latency | Latency | *Mbonye et al. (2018)* |
| HSPBP1 | – | Inhibit HIV-1 replication | Replication | *Chaudhary et al. (2016)* |
| HSPBP1 | – | Decreased by HIV-1 infection | Replication | *Iyer, Mitra & Mitra (2022)* |

Nastiti et al. (2024), *PeerJ*, DOI 10.7717/peerj.18002

**Table 4  Quality assessment of included studies.**

| Reference | Group I: test substance identification | Group II: test system characterization | Group III: study design description | Group IV: study results documentation | Group V: Plausibility of study design and data | Total | Reliability Category |
|---|---|---|---|---|---|---|---|
| Urano, Morikawa & Komano (2013); Urano, Morikawa & Komano (2013) | 3 | 3 | 3 | 3 | 2 | 14 | Reliable with restrictions |
| Joshi et al. (2013); Joshi et al. (2013) | 3 | 3 | 4 | 3 | 2 | 15 | Reliable without restrictions |
| Mercier et al. (2013); Mercier et al. (2013) | 3 | 3 | 5 | 2 | 2 | 15 | Reliable without restrictions |
| Anderson et al. (2014); Anderson et al. (2014) | 3 | 3 | 4 | 3 | 2 | 15 | Reliable without restrictions |
| Xia et al. (2015); Xia et al. (2015) | 3 | 3 | 6 | 3 | 2 | 17 | Reliable without restrictions |
| Chaudhary et al. (2016); Chaudhary et al. (2016) | 3 | 3 | 4 | 3 | 2 | 15 | Reliable without restrictions |
| Joshi, Maidji & Stoddart (2016); Joshi, Maidji & Stoddart (2016) | 3 | 3 | 6 | 3 | 2 | 17 | Reliable without restrictions |
| Painter, Zaikos & Collins (2017); Painter, Zaikos & Collins (2017) | 3 | 3 | 4 | 2 | 2 | 14 | Reliable with restrictions |
| Mbonye et al. (2018); Mbonye et al. (2018) | 3 | 3 | 4 | 2 | 2 | 14 | Reliable with restrictions |
| Ko et al. (2019); Ko et al. (2019) | 3 | 3 | 4 | 3 | 2 | 15 | Reliable without restrictions |
| Priyanka Wadhwa et al. (2020) | 3 | 3 | 5 | 3 | 2 | 16 | Reliable without restrictions |
| Chand, Iyer & Mitra (2021) | 3 | 3 | 5 | 3 | 2 | 16 | Reliable without restrictions |
| Iyer, Mitra & Mitra (2022) | 3 | 3 | 6 | 3 | 2 | 17 | Reliable without restrictions |
| Chand et al. (2023) | 3 | 3 | 5 | 3 | 2 | 16 | Reliable without restrictions |

isoforms. The MRJ-L was recognized for its interaction with HIV−1 Vpr, which promotes viral nuclear entry and replication (*Hanai & Mashima, 2003*). It has been established that the expression level of the MRJ-L exhibits a positive correlation with HIV-1 replication (*Ko et al., 2019*). Notably, *Chand et al. (2023)* conducted a study indicating that other specific isoforms of the HSP40 family, namely DNAJB3, DNAJB6, DNAJB8, and DNAJC5, interacts with the HIV-1 Vif protein, which causes degradation and consequently reduces HIV-1 infectivity. In contrast, most other isoforms studied, such as DNAJA3, DNAJB1, DNAJB7, DNAJC4, DNAJC5B, DNAJC5G, DNAJC6, DNAJC22, and DNAJC30, which bind to the same contact site, increase viral entry and replication, suggesting conflicting roles within the HSP40 family (*Chand et al., 2023*). These findings complement those of an earlier study by the same research group, which followed the progress of infection from the onset of viral infection, the time when the number of viruses peaks, to the decline phase. It showed that the majority of HSP40 isoforms respond to stress during the early to peak stages of HIV-1 infection (days 1–7), with the exception of DNAJB6a, which decreases during the peak stage. This fluctuation contrasts with certain HSP70 isoforms, where the expression of HSPA1A, HSPA1B, HSPA6 decreased on days 1–7, while other isoforms HSPA1L, HSPA2, HSPA5, HSPA9, HSPA12, and HSPA13 increased on and after day 7 (*Chand, Iyer & Mitra, 2021*). Additionally, during the early phases of HIV-1 infection, the former isoforms exhibit a significant increase, except DNAJB6, which later declined in the peak and late phase of infection. This fluctuation contrasts with certain HSP70 isoforms, where several are upregulated (*e.g.*, HSPA1L, HSPA2, HSPA5, HSPA9, HSPA12, and HSPA13), while others from the HSP70 family are downregulated post-peak infection phase (*Chand, Iyer & Mitra, 2021*). Furthermore, in neurons transfected with HIV-1 gp120 protein, HSP70 expression appeared to be decreased. Although the study did not specifically examine specific HSP70 isoforms, it supported the evidence that HIV-1 potentially suppresses HSP70 expression during the acute phase (*Xia et al., 2015*). Additional research on astrocytes infected with HIV-1 Tat has corroborated the results of this study. Mortalin, a specific member of the HSP70 family, exhibits a direct interaction with Tat, leading to its degradation. This interaction results in a reduction of Tat cytotoxic effects within the astrocytes (*Priyanka Wadhwa et al., 2020*). Further investigation is needed to characterize specific alterations in the expression of isoforms within the HSP40 and HSP70 families throughout the different phases of HIV-1 infection to identify potential biomarkers.

Another essential feature of the HSP40 and HSP70 is the ability of the two families to form complexes. The N-terminal J domain of HSP40 interacts with HSP70, which binds ATP and then catalyzes the hydrolysis of ATP to ADP. Client proteins bound to other domains of HSP40 are transferred to HSP70 and become substrates fixed by the structure of this chaperone complex (*Alderson, Kim & Markley, 2016*). Although not yet physically demonstrated, HIV-1 Rev is thought to be a client protein bound to one of the HSP40 domains. The resulting HSP40-HSP70 complex can inhibit the translation of the Rev protein, a regulatory protein that ensures viral RNA accumulation inside the host cell, thus preventing virus production (*Urano, Morikawa & Komano, 2013*).

HSP70 possesses an essential co-chaperone known as HSP70 binding protein 1 (HSPBP1), that serves as a nucleotide exchange factor, catalyzes ADP and ATP exchange

on HSP70, a key step in the chaperone cycle. In addition, HSPBP1 exhibits independent functions in regulating cellular proteins (*Mahboubi et al., 2020*). HSPBP1 also interacts with HSP70 and HSP40. Whether in conjunction with HSP70 or alone, HSPBP1 suppresses virus production mediated by HSP40. Moreover, independent HSPBP1 expression negatively correlates with LTR-driven gene expression in HIV-1 in the presence of Tat, affecting the nuclear factor pathway of kappa B (NF-κB) and reducing LTR interaction with p50/p65, thereby inhibiting viral transcription (*Chaudhary et al., 2016*). A decrease in HSPBP1 expression due to HIV-1 infection mediated by Tat through its binding to cellular gene promoters is identified. The downregulation of HSPBP1 enhances LTR-driven gene expression, promoting HIV-1 infection (*Iyer, Mitra & Mitra, 2022*). Based on these findings provides insight into the role of HSPBP1 as a suppressor of HIV-1 replication by inhibiting LTR-driven gene expression, either independently or as a co-chaperone. However, this suppressive effect can be counteracted by HIV-1 Tat, which suppresses HSPBP1 gene promoter activity (*Chaudhary et al., 2016*; *Iyer, Mitra & Mitra, 2022*).

The final HSP we examined is HSP90, which is a prominent member of the family. It roles multifaceted in different stages of HIV-1 infection, including viral transmission, uncoating, and latency reactivation. HSP90 has two different isoforms, HSP90α and HSP90β, expressed in HIV-1 microvesicles. Both isoforms stimulated the maturation of microvesicle-infected dendritic cells, with HSP90β being more potent than the former, and increased ICAM-1 expression, thereby facilitating HIV-1 transfer to T cells and enhancing the cell-to-cell spread of infection (*Mercier et al., 2013*). In another study, HSP90β/HSP90AB1 was found inside HIV-1 virions but outside the viral core. Here, HSP90AB1 inhibition rendered HIV-1 non-infectious, while both wild-type and capsid-mutated HIV-1 could infect target cells expressing HSP90AB1 (*Joshi et al., 2013*). This finding confirmed previous research, which showed the role of HSP90 as a host factor that supports HIV uncoating and rescues impaired HIV replication, making the virus more resistant to antiretroviral drugs (*Joshi & Stoddart, 2011*). In this case, HSP90, especially the HSP90β isoform, is thought to be a positive regulator of HIV-1 infection, which warrants further investigation.

In a latent state, HIV-1 remains inactive within cellular reservoirs without replicating or expressing viral proteins. Reactivation of latent HIV-1 occurs when infected cells are activated, which involves complex cellular and molecular mechanisms. In particular, HSP90 is a chaperone that interacts with specific molecular factors in this process (*Siliciano & Greene, 2011*; *Anderson et al., 2014*). HSP90 plays a critical role in activating the NF-κB pathway, particularly in activating the inhibitor of NF-κB kinase (IKK) complex and facilitating the nuclear translocation of RelA/p65. It is also indirectly involved in activating protein kinase C (PKC). Inhibition of HSP90 disrupts activation of the IKK complex and positively correlated with reduced HIV-1 reactivation from latency, suggesting that HSP90 acts as a chaperone facilitating viral resurgence (*Anderson et al., 2014*). In a separate study headed by *Joshi, Maidji & Stoddart (2016)*, it was found that blocking HSP activity effectively inhibits the transcription of HIV-1 within long-lasting viral reservoirs, thereby preventing the virus from reactivating.

Two independent studies present the relationship between HSP90 and HIV-1 transcription factor, the positive transcription elongation factor P-TEFb and its

components, cyclin-dependent kinase 9 (CDK9) and cyclin T1, which are required for reactivating latent provirus (*Painter, Zaikos & Collins, 2017*; *Mbonye et al., 2018*). HIV-1 Tat recruits P-TEFb to regulate transcriptional elongation by RNA polymerase I (*Asamitsu, Fujinaga & Okamoto, 2018*). HSP90 serves as a chaperone in the P-TEFb assembly. Inhibiting HSP90 leads to the degradation of cyclin T1, blocking P-TEFb formation and suppressing HIV-1 transcription. In HIV-1-infected T cells, the suppression of viral transcription reduces the reactivation of latent provirus (*Mbonye et al., 2018*). This finding aligns with other studies on HIV-1 latency in quiescent progenitor cells, where quiescent cells display minimal proliferation, differentiation, and cell cycle progression, ultimately increasing HIV-1 latency. In this context, inhibiting HSP90 enhances quiescence and latency. Additionally, it has been discovered that PKC stimulation is responsible for keeping latency in dormant cells, which aligns with findings by Anderson et al., suggesting the potential involvement of HSP90 in the PKC stimulation pathway to facilitate transcription during the reactivation of latent HIV-1 (*Anderson et al., 2014*; *Painter, Zaikos & Collins, 2017*). A summary of the HSPs involvement in HIV-1 infection and life cycle is presented in Fig. 2.

Nevertheless, there are several limitations to be considered in our review. Firstly, our analysis exclusively encompassed full-text articles in English, thereby omitting potentially valuable data from articles in other languages, locally published, or inaccessible articles, which may introduce evidence selection bias into our review. Secondly, there is no existing quality assessment tool that covers all the critical aspects of an *in vitro* systematic review, including the quality assessment tool we used, the ToxRTool, which is originally designed for the assessment of *in vitro* toxicological data. A recent systematic review also highlighted the need for a comprehensive review guideline and quality assessment tool for *in vitro* systematic reviews and meta-analyses (*Schneider et al., 2009*; *Tran et al., 2021*). However, most of the ToxRTool criteria are suitable to assess the reliability level of the studies and further substantiate our conclusions.

Our review systematically summarized the currently available literature on the interactions between HSPs and HIV-1 pathogenesis as an effort to approach the necessary steps in discovering new anti-HIV-1 strategies *via* host-virus interactions. Nevertheless, as all studies included in this review were conducted *in vitro*, the clinical implications remain to be determined. We hope that *in vivo* studies to validate the clinical significance of these interactions and their potential as targets for therapeutic intervention are within reach.

## CONCLUSIONS

This systematic review summarized the complex relationship between HSPs and HIV-1 infection, suggesting that these proteins have critical roles in the HIV-1 life cycle and anti-HIV-1 immune response. Further research is required to determine the clinical significance of this interaction and their potential as targets for future HIV-1 therapies.

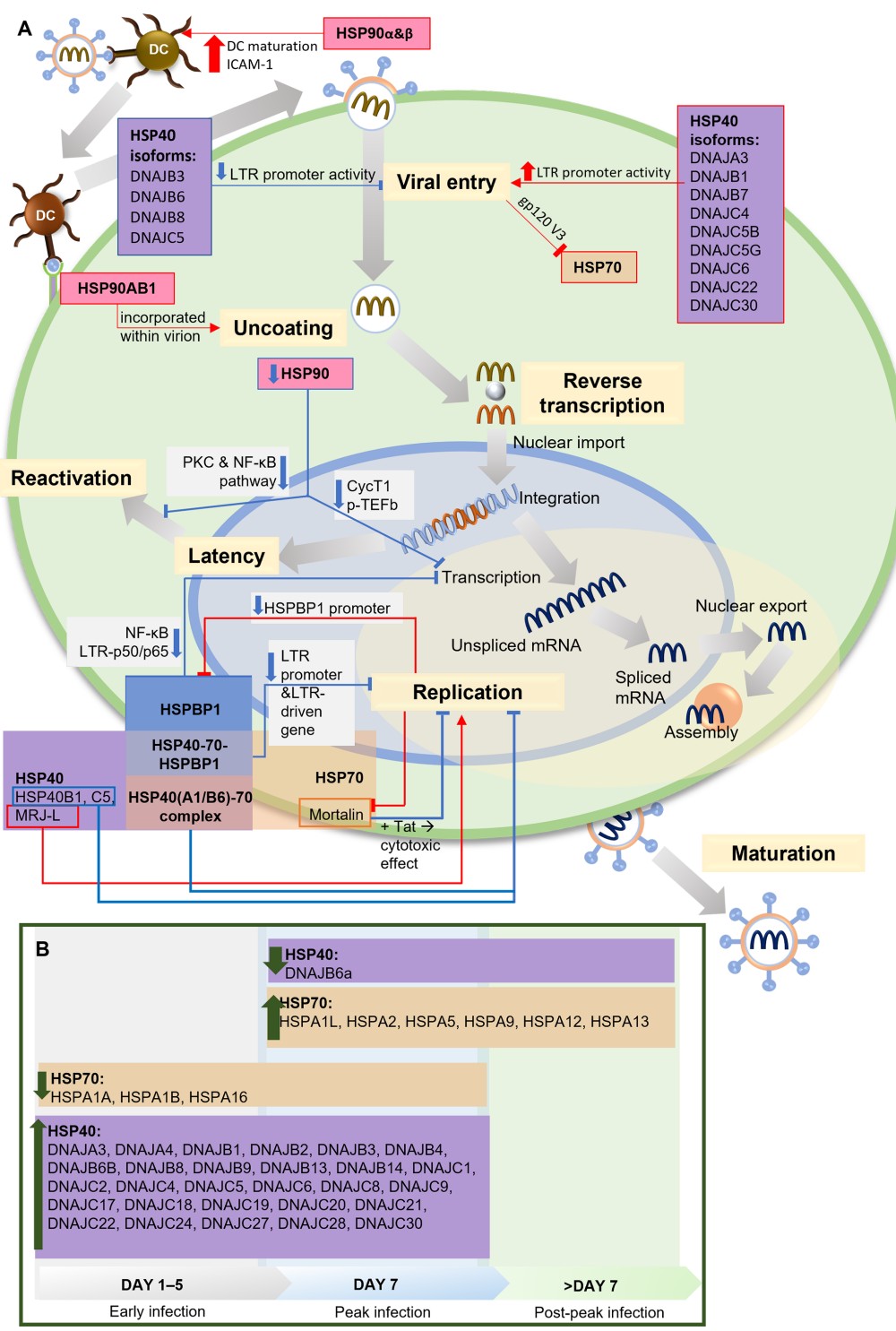

**Figure 2** **The role of HSPs in HIV-1 infection.** (A) The role and interaction of different HSPs in the HIV-1 life cycle. 

**Figure 2 (...continued)**
During viral entry, several HSP40 isoforms (DNAJB3, DNAJB6, DNAJB8, DNAJC5) inhibit the process by decreasing LTR promoter activity, while other isoforms increase LTR promoter activity. HSP90α & HSP90β increase DC maturation and ICAM-1 expression which facilitate the HIV-1 transmission from DC to T lymphocytes. HSP70 is inhibited by the presence of gp120 V3 during viral entry into cells. At the uncoating stage, HSP90AB1/HSP90β incorporates with virions to facilitate the process. In the replication stage, Mortalin, an isoform of HSP70, reduces the cytotoxic effect in the presence of Tat, thus inhibiting replication, as do some HSP40 isoforms (HSP40B1, HSP40C5), while another isoform (MRJ-L) supports replication. HSP40 and HSP70 form a complex that inhibits replication. HSPBP1 interaction with the HSP40-HSP70 complex reduces the activity of the LTR promoter and the LTR-driven gene, thereby inhibiting replication. HSPBP1 is known to inhibit LTR-p50/p65 interaction in NF-κB pathway, thus inhibiting mRNA transcription. Meanwhile, HSPBP1 expression itself can be inhibited because viral replication suppresses HSPBP1 promoter activity. HSP90 inhibition can inhibit CycT1-mediated p-TEFb formation, thereby inhibiting transcription, as well as inhibiting PKC and NF-κB pathway activity, thereby preventing viral reactivation from latency. (B) HSP40 and HSP70 isoforms modulation during HIV-1 infection. HSP40 and HSP70 isoforms show different modulation during the process of HIV-1 infection. Most of the HSP40 isoforms increased during early to peak stage, only one isoform (DNAJB6a) decreased during and after the peak stage. Several HSP70 isoforms (HSPA1A, HSPA1B, HSPA16) decreased during early to peak stage, while some others increased since peak stage. CycT1, Cyclin T1; DC, Dendritic Cell; gp120 V3, variable region 3 of HIV-1 envelope glycoprotein 120; HSP, Heat Shock Protein; HSPBP, HSP binding protein; ICAM-1, intercellular adhesion molecule-1; LTR, long terminal repeat sequence of HIV-1, MRJ-LMammalian relative of DnaJ-large isoform; NF-κB, Nuclear factor κB; p50/p65, protein 50/65 subunit in NF-κB pathway; PKC, protein kinase C; p-TEFb, positive transcription elongation factor b; Tat, HIV-1 Tat protein. Image generated using PowerPoint.

## ACKNOWLEDGEMENTS

Authors would like to express gratitude to Postgraduate School, Universitas Airlangga, for providing conducive academic environment which supported the completion of this study.

### Funding

Chyntia Tresna Nastiti received a scholarship from the Indonesia Endowment Fund for Education Agency (LPDP). The funders had no role in study design, data collection and analysis, decision to publish, or preparation of the manuscript.

### Grant Disclosures

The following grant information was disclosed by the authors:
Indonesia Endowment Fund for Education Agency (LPDP).

### Competing Interests

The authors declare there are no competing interests.

### Author Contributions

- Chyntia Tresna Nastiti performed the experiments, analyzed the data, prepared figures and/or tables, authored or reviewed drafts of the article, and approved the final draft.
- Noer Halimatus Syakdiyah conceived and designed the experiments, performed the experiments, analyzed the data, prepared figures and/or tables, authored or reviewed drafts of the article, and approved the final draft.

- R.M. Firzha Hawari analyzed the data, prepared figures and/or tables, and approved the final draft.
- Youdiil Ophinni analyzed the data, authored or reviewed drafts of the article, and approved the final draft.
- Ni Luh Ayu Megasari conceived and designed the experiments, performed the experiments, analyzed the data, prepared figures and/or tables, authored or reviewed drafts of the article, and approved the final draft.

## Data Availability

This is a systematic review/meta-analysis.

## Supplemental Information

Supplemental information for this article can be found online at http://dx.doi.org/10.7717/peerj.18002#supplemental-information.

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
