# Peer review of "The role of heat shock proteins in HIV-1 pathogenesis: a systematic review investigating HSPs-HIV-1 correlations and interactions"

_PeerJ, doi:10.7717/peerj.18002_

## Round 0.1 · original submission · Major Revisions

Overall, the manuscript "The role of heat shock proteins in HIV-1 pathogenesis: a systematic review" by Noer Halimatus Syakdiyah and colleagues presents a comprehensive overview of the involvement of heat shock proteins (HSPs) in HIV-1 pathogenesis. The reviewers have provided valuable feedback, highlighting both strengths and areas for improvement.

Reviewer 1 acknowledges the manuscript's clarity and execution but suggests adding relevant literature from 2024 and clarifying the data filtering process. They also inquire about the use of MeSH terms in the search strategy and point out a discrepancy in the number of studies analyzed.

Reviewer 2 praises the well-structured review but suggests deeper exploration of certain aspects, such as the specific mechanisms of HSPs in different stages of HIV-1 infection. They also suggest revising complex terminology and correcting the number of studies analyzed.

Reviewer 3 appreciates the paper's comprehensive summary but raises concerns about the focus on in vitro studies, the timeframe of the literature search, redundant information in the methods section, and unclear sections in the manuscript. They also suggest including visual depictions to enhance understanding.

In summary, the manuscript is well-written and executed but requires revisions to address the reviewers' comments. The authors should consider incorporating relevant literature from 2024, clarifying the data filtering process, revising complex terminology, and addressing the issues raised by Reviewer 3 regarding the focus on in vitro studies and the clarity of certain sections. Additionally, the authors may consider including visual depictions to enhance the presentation of their findings.

Reviewer 1 ·

Basic reporting

The manuscript is nicely written and executed.

Relevant Literature of 2024 should be added.

Results are well presented and discussed.

Experimental design

The data search strategy is clear and the protocol was previously recorded.

The filtering of data is fine except at one place in Figure-1: Out of 3062 article found, 253 were removed as duplicates. The reason of removing 2704 article is not clear from Figure 1 and its relevant text in Results

Validity of the findings

The results are discussed in detail in Discussion section. More strength could be provided by adding relevant references of 2024.

Additional comments

1. Have authors provided any MeSH terms to search as these terms are utmost clinical importance.
2. Line 224, ‘Quality assessment’ describes that 15 studies are analyzed whereas in Figure 1, fourteen studies are mentioned. Please correct it.

Reviewer 2 ·

Basic reporting

The review is well-structured, the introduction effectively sets the context for the study, explaining the importance of understanding the role of HSPs in HIV-1 pathogenesis. It demonstrates a thorough investigation of relevant literature, encompassing recent research developments. However, there could be a deeper exploration of certain aspects, such as the specific mechanisms through which HSPs influence different stages of HIV-1 infection in the Introduction section.

Experimental design

1. The writing style is generally clear and concise, making the content accessible to a broad audience. However, there are occasional instances of complex terminology that could be clarified or defined for readers less familiar with the subject matter eg . Line 47 and line 62-65. The authors are requested to revise the texts.

2 It was mentioned in the Line 224 (Quality Assessment)-that 11 studies out of 15 analyzed studies have ……… and relevant to their intended purpose. The authors are requested to revise this line as only 14 studies were chosen for the analysis not the 15.

Validity of the findings

1. It would be better if the authors provide the final conclusive diagram of the outcome of the study for better clarification as the above mentioned study is too vast and involved different HSPs.
2. The authors stated in the limitation portion of the study that they employed the ToxRTool checklist as our primary measure to evaluate the quality of in vitro studies which is not specifically designed to evaluate the quality of in vitro studies. Is it possible to use another tool for the analysis?

Additional comments

The review provides a commendable analysis and synthesis of the reviewed literature, highlighting key findings and elucidating potential mechanisms underlying the relationship between HSPs and HIV-1 pathogenesis. However, meta-analysis of study could strengthen the review and would be an add on to the study.
Overall, the systematic review provides a comprehensive overview of the role of HSPs in HIV-1 pathogenesis, offering valuable insights and recommendations for further research. Addressing minor issues related to clarity, evidence presentation, and argumentation could enhance the impact and readability of the review.

Annotated reviews are not available for download in order to protect the identity of reviewers who chose to remain anonymous.

Reviewer 3 ·

Basic reporting

no comment

Experimental design

no comment

Validity of the findings

no comment

Additional comments

The manuscript titled "The role of heat shock proteins in HIV-1 pathogenesis: a systematic review" by Noer Halimatus Syakdiyah and colleagues provides a comprehensive summary of the involvement of heat shock proteins (HSPs) in the pathogenesis of HIV-1 infection. HIV remains an incurable disease, so any direction taken in this area would be of considerable value. Although the paper is intriguing, several issues need to be addressed:
1) It is important to point out that the authors did emphasize the role of HSPs in HIV pathogenesis, but why did they focus solely on in vitro studies and not on in vivo studies?
2) Why is this review restricted to studies from January 2013 to December 2023?
3) There is a significant amount of redundant information in this research, particularly in the Materials & Methods section. The details on inclusion and exclusion criteria, study selection, and data extraction appear to be repetitious. Additionally, the subsequent portion is difficult to comprehend. Could the authors please clarify the scientific criteria for the portion between 137 and 138?
4) In sections 143-144, it appears to be repeating itself or maybe such information can be compiled in the authors' contributions separately.
5) It is unclear why only limited country studies are included in the characteristics section.
6) Line 195 does not indicate which study the authors are referring to.
7) There is a possibility that the authors can merge the Role of HSP in HIV infection pathogenesis to the heading HSP and HIV-1 correlations, together with an appropriate title.
8) The authors should take serious note of the redundancy of contents since quality assessment appears again in line 223. Since they've already discussed it before, what's the reason for rehashing it? Further, this whole section is difficult to follow.
9) There is something unclear about lines 62-64. The same applies to lines 249-253.
10) Considering that ToxRTool is not designed for in-vitro studies, I wonder why the authors are employed (see statements 335-336).
11) Despite the authors' emphasis on Heat Shock Proteins (HSPs) in relation to HIV pathogenesis, there is a dearth of visual depictions that effectively demonstrate the significance and their possible consequences in HIV pathogenesis. The authors have the option to incorporate figures or graphics to enhance the visual depiction.

---

## Round 0.2 · accepted · Accept

Based on the thorough reviews and positive feedback from both reviewers, the manuscript has undergone significant improvements and now meets the journal’s standards for publication. Reviewer 2 has noted substantial elaboration on the mechanism of HSPs and their role in HIV-1 pathogenesis, corrections in the Results section, and a clear and impactful summary of findings. Reviewer 3 has confirmed the clarity and conciseness of the manuscript’s data search strategies and detailed discussion of the results. Both reviewers recommend final acceptance of the manuscript.

I concur with the reviewers' recommendations and recommend that the manuscript be accepted for publication.

Reviewer 2 ·

Basic reporting

1. The authors have elaborated the mechanism of interaction between HSPs and HIV-1 on different stage of the viral life cycle in Introduction which now explains the importance of understanding the role of HSPs in HIV-1 pathogenesis.

Experimental design

1. The authors have made adjustments to the designated texts.
2. The necessary correction has been made in the Results section; Quality of the included studies sub-section) as per the suggestion.

Validity of the findings

1. The authors have have summarized the involvement of HSPs in HIV-1 infection and life cycle in the provided Figure nicely could now enhance the impact and involvement of HSPs and readability of the review.
2. The explanation provided by authors regarding the use of ToxRTool checklist to evaluate the quality of in vitro studies seems to be fine.

Additional comments

The authors have made significant improvements and have adequately addressed all the points raised in the initial review. I recommend final acceptance of the manuscript as it now meets the journal’s standards for publication.

Annotated reviews are not available for download in order to protect the identity of reviewers who chose to remain anonymous.

Reviewer 3 ·

Basic reporting

The manuscript is well written.

Experimental design

Data search strategies are concise and clear.

Validity of the findings

There is a detailed discussion of the results.

Additional comments

None